# Review on the Integration of Microelectronics for E-Textile

**DOI:** 10.3390/ma14175113

**Published:** 2021-09-06

**Authors:** Abdella Ahmmed Simegnaw, Benny Malengier, Gideon Rotich, Melkie Getnet Tadesse, Lieva Van Langenhove

**Affiliations:** 1Department of Materials, Textiles and Chemical Engineering, Ghent University, 9000 Gent, Belgium; Benny.Malengier@UGent.be (B.M.); rotichgideon2016@gmail.com (G.R.); lieva.vanlangenhove@ugent.be (L.V.L.); 2Ethiopian Institute of Textile and Fashion Technologies, Bahir Dar University, Bahir Dar 6000, Ethiopia; melkiegetnet23@gmail.com; 3Clothing and Textile, School of Engineering and Technology, South Eastern Kenya University, Kwa Vonza 90215, Kenya

**Keywords:** microelectronics, e-textile, smart textile, interconnection, textile-adapted

## Abstract

Modern electronic textiles are moving towards flexible wearable textiles, so-called e-textiles that have micro-electronic elements embedded onto the textile fabric that can be used for varied classes of functionalities. There are different methods of integrating rigid microelectronic components into/onto textiles for the development of smart textiles, which include, but are not limited to, physical, mechanical, and chemical approaches. The integration systems must satisfy being flexible, lightweight, stretchable, and washable to offer a superior usability, comfortability, and non-intrusiveness. Furthermore, the resulting wearable garment needs to be breathable. In this review work, three levels of integration of the microelectronics into/onto the textile structures are discussed, the textile-adapted, the textile-integrated, and the textile-based integration. The textile-integrated and the textile-adapted e-textiles have failed to efficiently meet being flexible and washable. To overcome the above problems, researchers studied the integration of microelectronics into/onto textile at fiber or yarn level applying various mechanisms. Hence, a new method of integration, textile-based, has risen to the challenge due to the flexibility and washability advantages of the ultimate product. In general, the aim of this review is to provide a complete overview of the different interconnection methods of electronic components into/onto textile substrate.

## 1. Introduction

The term “smart textiles” is coined to designate an intelligent textile material and it covers a broad range of textiles. Smart textiles can be expressed as textile materials that are able to sense and respond to changes in their environment. Broadly, smart textiles are divided into passive and active smart textiles. Passive only observes, while active senses and reacts to the environmental changes [1,2].

Smart and wearable textiles are complete body-borne electronic systems with various functionalities of sensors which detect pressure [3,4], temperature and humidity [5,6], strain [7], chemical and bio-sensors [8,9], data processing and networking [10], mechanical actuation based on shape memory materials or electro-active polymers [11], thermal and energy generation [12,13], as well as energy storage [14,15] and smart fashion [16,17]. These textiles also contribute to help communication such as health surveillance, safety, comfort, and leisure [18].

Scholars have tried to develop wearable computers to simplify our lives, however, the challenge was how to incorporate computer hardware unobtrusively into/onto clothing materials [19]. Common hardware is not comfortable enough during wearing due to its rigidity, high weight, and other discomfort qualities. Thorp [20] stated that the first wearable electronic computer was developed in 1955 by Claude Shannon to calculate roulette probabilities. Then, Thorp and his co-workers developed wearable computers in 1961 by using switches in the shoe for input, acoustic output through a tiny earplug, and a small handmade computing unit worn on the belt to calculate roulette probabilities [21]. The first wearable computer systematically produced came out in 1968, when Ivan Sutherland presented a head-mounted display using small cathode ray tube (CRT) displays placed in front of the user’s eyes [22]. One of the pioneers in this field, Steve Mann, developed several prototypes using a near-eye display, an on-body computer, and a one-handed input devices [23].

Ojuroye [24] studied the design of a novel classification chart that measures the extent of electronic integration within textiles and this chart indicated how the level of electronic integration in textile impacts the degree of functionality, commercial viability, and industrial compatibility. Rambausek et al. [25] discussed the five different levels of textile integration and its technical challenges per integration level. Bosowski et al. [26] generally studied the three category levels of the integration of electronic components and circuits into/onto textiles which can be distinguished as textile-adapted, textile-integrated, and textile-based, as shown in Figure 1.

The first type, the textile-adapted, refers to the manufacturing of special clothing accessories or extensions to contain electronic devices, so the electronic function is integrated by adding the actual rigid electronic device onto or into the garment (e.g., ICD+, communication devices for firemen and MP3 players) [27,28,29]. The clothing and electronics are processed separately and merged at a later stage. Typically, the electronics can again be removed to allow washing and maintenance of the clothes, as shown in Figure 1a. However, the textile being subjected to multiple mechanical deformations such as stretching and related characters. This puts lots of tension on any present interconnections via the textile between components and leads to lower reliability of the interconnection part of electronic device.

In the second type, the textile-integrated, the integration of the electronic components is done through the creation of an interconnection between the electronic elements and textile substrate, and often the textile material performs selected functionalities (e.g., a metal push buttons as on/off switch). Integration is at the fabric level, with circuits that can be partially fixed on the surface of/or inside the fabric [30]. Here, conductive fiber interconnections have been sewn, stitched, printed, embroidered, and soldered into the fabric as shown in Figure 1b. This kind of integration attains some level of flexibility and even stretchability, and is already more user friendly than textile-adapted solutions.

The last type of integration of electronic components is the textile-based which uses the textile structure itself to provide higher end electrical components within an electrical circuit embedded within the yarn or fiber. At the moment, these are mainly evidenced in research and in patents. No commercial products are out yet, but examples include in reference [30] (e.g., electro-conductive or metallic-coated multifilament yarns forming an antenna, or a thermopile couple from nickel-coated carbon fibers to generate electricity [31] (see Figure 1c), and an electrochemical transistor (ECT) integrated into silk fiber) [32].

In addition, a the European committee of standardization, within the technical committee working group of 31(WG31) was [33] summaries the level of integration of electronics components or device into or onto a textile product by four different level as described below.

The first level of integration is level 1 (removable): electronic device is added in a way that it is removable without destroying the product, e.g., by means of a pocket, touch-and-close fastener, push button. Second level of interconnection of electronics (Integration level 2 also called attached), in which an electronic device is attached to textile in a way that it is not removable without destroying the product, e.g., by means of stitching, welding, or gluing the device onto the textile. The third level of interconnection (or integration level 3) is known as mixed solution. Electronic device consists of one or more components made of textiles or through textile finishing and combined with permanently or non-permanently attached electronic components, e.g., an LED lamp attached to a conductive track woven into a fabric. In addition, the last level of interconnection (or integration level 4) is known as full textile solution. All components of the electronic device are made of textile or textile finishing.

The higher the degree of integration, the more standard manufacturing processes of textiles and electronics are used, to be revisited and adapted. Development of the integration technologies is crucial for wearable systems to meet the required comfort and reliability. Flexibility, comfort, and lightweight properties are maintained at its highest level with textile-based integration.

Smart textile encompasses any fiber, yarn, flat, or 3D textile product that changes its properties under a change of functional value in some physical quantity or stimulus. In this way, smart textile allows us to achieve an active realization of textile functions i.e., protection or integrated lighting intended for interior decoration [34]. By integrating programmable wearable electronics into/onto textiles substrate, the textile material became an intelligent textile and could fulfil more and more complex functions such as active protection and monitoring of human health [35]. The limitation on user comfort and the low flexibility of this increasing number of portable electronic systems must, however, be kept to a minimum. Preferably, these systems should be quasi-invisible and non-noticeable to the user [36].

At the end of the 20th century, further adaptation of the electronic-textiles was a fundamental step towards growth. Most parts of the wearable electronics are replaced by textile-based electronics. That means there are textiles containing electronic components and parts of the textile are conductive, with the aim of keeping a textile aspect or feel as much as possible. Parallel to this development, a large minimization of electronic component size occurred due to joint advances in material science and electronics, which led to expansion of the potential of embedding electronics within clothing. A main breakthrough regarding the development of e-textiles was the discovery of conductive polymers. Conductive polymers invented by Shirakawa et al. [37] turned out to be a key innovation for conductive textiles. Many inventions, innovations, and patents on the development of wearable technology, for the application of wearable textiles, build upon conductive polymers [38,39] such as smart nanotechnology and integration of interactive electronics onto textile substrate [40,41].

The integration of electronic devices onto textile materials has its limitations. For instance, attaching or detaching electronic components to or from conductors embedded onto the textile is difficult to achieve. The textile material will undergo manipulation such as stretching, therefore the conductors must move and stretch with the material [42,43]. In addition, the integration of conductive material leads static electricity due to tribo-phenomenon or due to induction charging [44]. Static electricity arises when surfaces that were in contact are separated. The rubbing of two insulating materials against each other can give rise to a static buildup of up to several thousand volts. The build-up charge is then suddenly released, either by a spark or a surge, when a certain voltage is reached; the latter is influenced by several factors such as the conductivity of the charged body or the humidity of the environment [45,46,47]. In addition, humidity can play a role as well: humidity is relatively high when clothes are worn. This can destroy sensitive components of the electronics. Insulating conductive trucks and electronic components with nonconductive polymeric material are able to dissipate static charges even without grounding [48]. Anti-static properties can be conferred by coating fibers with metallized films, intrinsically conductive polymers, polyelectrolytes, or by low molecular weight anti-statics in solution [49].

The integration of electronics into textile must take this into account. Diverse technologies have been developed and investigated to create the interconnections of electronics with textile substrates. Each one of them has its benefits and disadvantages. The integration of electronics into textile is supposed to have limited or no effect on the comfort, flexibility, fashion statement, wearability, and ergonomics of the garment [50,51,52,53].

According to De Mulatier et al. [54] there are four different development stages to smart textile fabrication: flexible circuits, hybrid stretchable circuits, textile circuits, and functionalized fibers. At the early-stage development of e-textile, basic textile circuit technologies are involved by attaching standard electronic components directly onto fabric to create the textile circuits. As the next stages, textile integrated flexible circuits and hybrid stretchable circuits were the emerging technologies, the fabric serves as a substrate for the electronic circuit to maintain essential textile properties. While in the future, all the electronics might be embedded as functionalized fibers. In this fourth development stage, the functionalized fibers will have replaced the rigid electronic parts. To make the electronics even more integrated, functionality shall be diffused inside the core of the fiber material. However, the recent development is limited to basic logic circuitry and quasi-static applications.

According to Li et al. [55] three dimensional deformable, highly stretchable, permeable, durable and washable fabric circuit boards embedded in textile have the advantage to rely on more established process technologies with a dedicated substrate separately processed from the garment. The substrate can be a mixture of rigid, flexible, and stretchable parts and must later be integrated into the garment substrate. In this review, the various types of integrating microelectronic devices into/onto wearable e-textile are discussed in detail. Special focus is on the level of the textile-integrated smart textiles, to obtain flexible circuits or hybrid stretchable circuits in the textile through the combination of a textile-based circuit and a common microelectronic part as found on rigid circuit boards.

## 2. Integration Techniques

At the early-stage developments of smart textile, technologies involved integration of electronics directly into/onto textile materials, this by applying different connection approaches between the rigid electronic components and soft textiles. In order to interconnect different electronics components such as conductive yarns, sensors, batteries, and processing circuits, the conductive tracks have to embed directly into/onto the textile. The most basic function of the chip-level interconnections is to provide electrical paths to and from the substrate for power and signal distribution.

The integration of electronics into/onto the textiles requires two straightforward connection steps. The first one is the mechanical connection to a textile material, while the second step is the electrical connection integrated on the conductive structures. Both connections must be functional and reliable.

### 2.1. Mechanical Connectors

In 1964, the first scientific paper concerning the mechanics of pressure connections was published by Whitley [56]. The author gave a good overview and described the fundamental processes involved in crimping and force fit. In 1995, Mroczkowsk [57] proposed a cold welding process for crimped connectors from Ag and Cu wire with a barrel (brass) surface that had large-area metal-to-metal contact, and would hold it there under any desired environmental condition for a long lifetime. To detect the extensive deformation of the conductors, the surface has been analyzed using SEM. Bernardoni et al. [58] developed a low profile mechanical interconnect system having metalized loops and hoops for the creation of electronic connections. These crimp connections, as well as a good overview of crimping in general, has been published by Mroczkowski [59].

A study on the reliability of crimp connections was published in 1978 containing a mathematical model and the factors which affect the reliability of crimp connections [60] Optimization of the tool design, determination of parameters, effect of friction and comparison of implicit and explicit finite element methods to model crimp connections have been further investigated by different scholars [61,62]. Simon et al. [63] worked on the development of a multi-terminal crimp package for smart textile integration based on Crimp Flat Pack (CFP) which is a lead-frame-based electronic package that features crimp terminals for integrating electronics into textiles.

In addition to crimp beads, Kalhnayer et al. [64] developed a mechanism of connection for antennas on textile substrates by the integration of the transponder through an interposer between chip and fabric. Although the processes seem suitable, it was not possible to achieve a permanent electrical contact and higher deformation of the antenna occurred.

Researchers in [65,66,67,68,69,70] described various approaches of physical attachments for electronic PCB (Printable Circuit Board) and wire with a textile by using different methods such as snap buttons, socket buttons, bolt connection, and ribbon cable connector, as shown in Figure 2.

In 2014, Seager and Chauraya [71], studied the use of conductive hook and loop flexible and detachable connectors as the connector between the traditional electronics and a fabric system, and demonstrated how the electronics can be removed from a fabric system for security or other reasons such as washing. In 2015, Berglund and Duvall [72], and Molla et al. [73], introduced a novel technique for assembling surface-mount fabric PCBs using stitched traces and reflow soldering techniques. The results showed that all configurations were sufficiently durable for low-intensity wear, while, for high intensity wear, larger components and traces are needed to improve the durability.

Rubacha and Zięba [74], developed and studied multifunctional magnetic fiber which were manufactured by introducing ferromagnetic nano-particle powders into the fiber matter during fiber production for use in Textronic products where the magnetic fibers may be used for the construction of textile magnetic coils. Scheulen et al. [75] used adhesive bonded neodymium magnets for contacts in smart textiles. The electrical contact resistance between two magnets was found to be less than 0.01 Ohm. The magnets were glued to the textile using a conductive epoxy adhesive.

Righetti et al. [76] integrated electronics onto fabric through gold-coated neodymium magnets which then bonded to the fabric through a commercial cyanoacrylate adhesive (CA). In this way, the authors developed a modular I2C-based wearable architecture where the garment provides the I2C bus made of four conductive wires. At different positions, modules were attached to the bus via magnetic connectors.

### 2.2. Soldering

Soldering interconnections between microelectronics and conductive textile material is a process in which two or more items are joined by melting a filler metal or solder and putting it into the joint [77]. To be functional, the filler metal must have a lower melting point than the conductive thread or fabric and the microelectronics [78]. Soldering is normally done above 200 °C, although low-temperature solder, with melting points as low as 150 °C [79] or below, exist [80,81] and start to be common for textile integration.

Soldering involves mounting the components directly onto the textiles surface. It is necessary to transfer thermal energy from a heat source to the soldering point to melt and flow the solder between terminals of the microelectronic component (known as solder pads) and conductive thread(s) in order to create good connections. In the soldering process, heat can be transferred either by conduction, convection, or radiation [82]. In soldering, an additional material, called solder is used. The solders are soft alloys of lead (Pb), tin (Sn), or sometimes silver (Ag) that are used to join the metallic electrical components with the textile substrate, with Sn42Bi 58, a suitable alloy for textiles with melting point at 138 °C [83]. Soldering achieves good electrical contact [84]. The integration of the electronics on to textile substrate by soldering can usually be achieved by direct contact or frictional soldering [85], hot air or thermal soldering [86], ultrasonic soldering [87,88], laser soldering [89], and infrared soldering [90] as shown in Figure 3 [91].

Mostly, for textile materials, thermal bonding and ultrasonic soldering have been used. The advantage of friction bonding compared to thermal bonding is that the heat is generated directly at the joint. Localized heating reduces the risk of burning the surrounding nonconductive fabric material [92].

Additionally, the soldering of smart textiles was investigated by different researchers such as Molla et al. [73] who integrated LEDs into textile structures by soldering conductive uninsulated yarns. The LEDs were soldered to the exposed conductors, but in particular, the solder wicking along the multi-filament affected the durability of the joint and added stiffness to the textile. In addition, Mikkonen and Pouta [93] studied the integration of electronics onto conductive wire by direct soldering and insertion into two layer weaving fabric. The distinct effect of the solder, especially, has an impact on the stiffness of the textile. All work steps were performed manually and were not fast enough. Overall, the contacting processes, manually and automatically, were still too slow and the conductors were not stretchable.

The developments of woven electrical circuits on to fabric using several microchips connected after copper conductors have been woven into a fabric connected by using resistive welding (using top-bottom probes and parallel gap welding) were formed at the cross-over points of orthogonal conductive yarns was studied by Dhawan [94]. Furthermore, Atalay [95] developed textile-based transmission lines using stainless steel and silver-plated conductive yarns, which were inserted between two layers of navy blue colored polyester fabric structure, without any undulation in a straight line via applying ultrasonic welding technology for e-textile applications. The results show that stainless steel yarns were simply affected in terms of conductivity change. The silver-plated yarns were found to be less suitable for ultrasonic welding technology. However, silver-plated yarns with higher linear density showed satisfactory results at moderate working conditions.

### 2.3. Sewing and Embroidering

Sewing and embroidery technologies are methods for interconnection of electronic components by attaching them on top of a textile fabric with yarn. T. Linz et al. [96] stated that these methods are conventional techniques which consists of chip elements, PCB, and sensors that are placed on the substrate and attached by sewing to provide a rigid mechanical connection between the chip elements and conductive fabrics.

Temporary contacts and embroidered circuitry with conductive yarn has been demonstrated by Linz [97]. The work focused on the interconnection process and studied machine embroidered electrical contacts. They suggested a solution to improve the reliability by using a conductive yarn embroidered through a metalized contact area on a flexible substrate which produced an electrical and mechanical connection.

Different authors, such as Linz and Christine [98] and Hamdan and Voelker [99], showed that common stitch embroidery technology can be used to integrate electronics on to textile in a light and cost efficient way. Embroidering mechanisms were used for flexible electronic modules using flexible conductive yarn interconnection with sensors, batteries, textile keyboards, and electrochemical biosensors [100,101].

Sahta et al. [102] developed textile-based sewn switches by using commercially available conductive thread based on silver-coated multifilament polyamide thread and often used to connect devices within e-textile structures. Silver-coated multifilament polyamide yarns are elastic, flexible, and with respect to their textile properties, correspond to the “conventional” textile yarns. They are suitable for integration into the fabric structure by sewing. However, the thread used with a sewing machine as needle thread suitable are only 110dtex/f34x2-ply twisted yarns. Furthermore, the effects of washing on electrical conductivity of silver coated polyamide yarn was highly intensive [103].

Post and Orth [104] developed e-broidery (electronic embroidery in which the patterning of the conductive textiles is controlled by numerical sewing or weaving processes) as a means of creating computationally active textiles. The embroidery needle stitched through metalized contact areas of the substrate thereby making a mechanical and electrical interconnection with the conductive thread. The analysis of embroidery contacts for electronics in textile and investigation of ways to improve reliability of the embroidery contact on the textile have also been studied in [105]. Christine et al. [106] developed a fully integrated electrocardiogram shirt based on embroidered electrical interconnections with conductive yarn and miniaturized flexible electronics. Sewn electrode onto textile substrate using conductive stainless steel and polyester composite threads is shown in Figure 4 [107].

Today, researchers have developed advanced embroidery machines to integrate electronics into the textile substrate. Advancements of Textile Research Facility TITV, from Greiz in Germany [26], invented an embroidery technology using luminous sequins embroidered to conducting yarn, which is available on embroidery machine stich file (ZSK) embroidery machines with a ZSK sequin device. The sequin feeder apparatus includes at least two sequin feed units, each including a sequin feed mechanism, for feeding a continuous sequin strip toward a predetermined cutting position and a sequin-cutting cutter section located in the predetermined cutting position as shown in Figure 5. One of the sequin feed units is selected and positioned in a predetermined sewing operation position, and the driving force of a feeding drive mechanism is transmitted to the sequin feed mechanism to feed out and integrate the sensor, LED sequins, and antenna on textile fabric. However, when embroidery is used, especially in near-body applications, the electronic and conductive paths have to be placed on the outside of the fabric, as they generally should be shielded from naturally occurring conductive substances such as sweat [108]. Furthermore, the electronic components and conductive truck must not be damaged or lose efficiency by any external interruption such as abrasion and cyclic washing. Thermoplastic polyurethane was laminated over the embroidered conductive yarn to successfully protect it during washing. Applying encapsulation by using epoxy compound transfer, molding, adhesive film protection, and hot melt encapsulation to protect electric contacts is very common practice. It protects against mechanical stress, including temperature and induced mechanical stress. It also protects against chemicals such as moisture, salts, and atmospheric contaminants such as sulfur oxides (SO2) [97].

The main attention point with embroidery and sewing is that special yarn, and even needles, must be used, which allow embroidering without yarn breakage. Special yarns have been developed, e.g., Madeira high conductive thread (HC12) [109], though further improvement in these yarns is still needed to obtain a wider range of conductivity, strength, and abrasion resistance. In addition, Silver-Tech 50 [110], Nickel Conductive yarn [111], Volt Smart Yarns, or CleverTex^®^ [112] threads are also suitable for high resolution embroidery and have high resistance to washing cycle, which were used as a smart solution for wearable applications.

### 2.4. Hybrid Solder and Sewing Integration

In 2015, Bergluned et al. [72] integrated microelectronics in the fabric by using a Brother PR650e embroidery machine in a hybrid solder and sewing technique. Three surface-mount LED packages (sizes 1 mm, 3 mm, and 5 mm) were attached to stitched conductive traces on 100% cotton fabric with a plain weave structure in two configurations, parallel with, and perpendicular to, the package axis, as shown in Figure 6. The conductive traces were created using Syscom Liberator 40 silver coated Kururay^®^ Vectran conductive yarn, stitched in a lockstitch structure, using the embroidery machine and components, were attached manually using a reflow soldering process, with low-melt solder paste (Chip Quik).

The durability test was performed, using simulated every day wear testing, and showed that the surface-mount soldering of components to stitch conductors was a viable method of fabricating e-textile circuits. However, a challenge was excessive backstitching over the traces, extraneous or redundant stitched and thread trimming. These problems led to risk of electrical shorts, and the washing of the e-textile was found to be unsatisfactory.

### 2.5. Electrical Conductive Adhesive

Bonding involves applying conductive adhesives to embed components into/onto textile substrates. Conductive adhesives may be developed according to the end use application. Non-toxic, highly conductive, highly durable, and moderately flexible conductive adhesives can potentially be used to bond rigid components with flexible textile substrates [113]. Joining also involves attaching an electronic component to a fabric by cyanoacrylate adhesive and applying (magnetic) force or heat on the fabric [114,115], as shown Figure 7.

Conductive adhesives are widely used in the electronic packaging applications, such as die attach and solderless interconnections. For this purpose, different types of conductive adhesives for electronics packaging have been developed [116,117,118]. Snacaktar et al. [119] and Mehmann et al. [120] have studied epoxy-based adhesives used for joining electronics parts to fabric circuits. Krshiwoblozki et al. [115] described electronic circuits connected with fabric by using thermoplastic polyurethanes nonconductive adhesive bonding. The thermoplastic adhesive bonding technology for bonding of electronic modules onto the textile substrates uses a thermoplastic nonconductive adhesive (NCA) film. The modules are placed onto textile circuits, with an NCA film in between, by applying pressure and heat. The adhesive melts and contact partners touch, as shown in Figure 7 and Figure 8.

Unlike other types of adhesives, electrically conductive adhesives perform two primary functions. First, conductive adhesives form joints with sufficient strength so that they can bind two surfaces, and secondly, an electrical interconnection was formed between the two bonded surfaces. This dual functionality is usually achieved in composite form by dispersion of particles in an insulating adhesive matrix [116] by controlling the process parameter such as temperature, pressure, cure time, pot and shelf life, which are critical to the success of making reliable electrical and mechanical interconnections [121]. Li and Wong [122] applied conductive adhesives as a lead-free alternative in electronic packaging and interconnecting electronic material and in flip-chip assembly, chip scale package (CSP), and ball grid array (BGA) applications in replacement of solder. The recognized advantages of using electrical conductive adhesive ECA are improved environmental impact, gentler processing conditions (allowing the use of heat-sensitive and low-cost components and substrates), and fewer processing steps (reducing processing cost). However, this type of technology is not available easily and has challenging issues such as lower electrical conductivity, low conductivity fatigue resistance, limited current-carrying capability, and poor impact strength.

The conductive adhesive methods of joining electronics to conductive textiles is confronted with limitations including lower conductivity than solder methods, sensitivity to the type and quality of component and board metallization, requiring longer time to cure, and possessing lower durability in various climatic environments [123]. One issue is thermal stress caused by coefficient of thermal expansion mismatch between the conductive textile and electrical component, while other attention points are mismatched between the adhesive and adherent during temperature cycling, oxidation of the bonding surfaces and of the filler, and degradation by UV-light or corrosive gases [124,125,126].

### 2.6. Inkjet and 2D Screen-Printing

Another method for practical integration of electronics onto textiles substrate is printing textile-based electronics onto the surface of the substrate. This can be performed by primary printing techniques, such as screen printing or inkjet printing [127]. Examples of smart textiles achieved by printing techniques are piezo resistive layers [128], a textile transmission line [129], heating device [130,131], a frequency selective surface [132], a secured traceability tag [133], an auto touch pad [134,135], highly deformable electro chromic device and electrochromic e-skin [136,137,138], organic light-emitting diodes [139,140,141], microchips such as RFID tags [142,143,144], resistors and capacitor [145] flexible heated circuits [146,147], antenna [148,149,150], and piezo sensitive materials [151].

Today, various electro active functional inks allow the manufacturing of textile based electronic devices. Several researchers have developed printable conductive inks for screen printing that can form interconnections [152,153,154,155]. They showed that these wearable electronics were rigid and inflexible electronic technologies that offer limited skin-compatibility and are damaged under washing. The resultant e-textiles were too uncomfortable to wear because they were not breathable.

In 2017, Carey et al. [156] developed an ink based on graphene and other two-dimensional materials that enable the printing of washable and biocompatible electronics on textiles such as cotton and polyester. Cao et al. [157] also developed washable, breathable, and designable electrodes through screen-print carbon nanotube (CNT) ink, which not only had excellent flexibility and stability but also relatively low surface resistivity (0.2 kΩ/sq.) and air permeability (88.2 mm/s).

Paul et al. [158] developed an innovative washable screen-printed network of electrodes associated with conductive tracks on textiles for medical applications as shown in Figure 9. A polyurethane paste was screen printed onto a woven textile to create a smooth and high surface energy interface layer. Subsequently, a silver paste was printed on top of this interface layer to provide a conductive track and a final polyurethane encapsulation layer was placed to protect the conductive track from abrasion and creasing.

Researchers [159,160] have also overcome the limitations of washability by developing low-boiling point inks based on nano scale platelets of graphene and hexagonal boron nitride (h-BN) suspended in organic solvents that were compatible with inkjet printers. The researchers printed stacks of inks based on different two-dimensional materials onto cotton and polyester fabrics to create electronic components such as amplifiers, programmable memories, logic circuits, and integrated circuits, as illustrated in Figure 10. The papers showed that one of the problems with current approaches to inkjet printing onto a two-dimensional material is that commonly used solvents have a low-boiling point and are toxic.

### 2.7. Three-Dimensional (3D) Printing

Additive Manufacturing (such as 3D printing) is a form of manufacturing in which a Computer Aided Design (CAD) model is captured and then subsequently fabricated in a layer-by-layer manner [161]. The first printer to print electronic elements was sold in 1936. It was an incredible revolution that brought change in the way we conceive electronic materials [162].

An advanced example of using 3D printing for the integration of electronic components has been given by Grimmelsmann et al. [163]. For the 3D printing, an Orcabot XXL 3D printer was employed, which works with the principle of the fused deposition method (FDM) technology. In this technique, a filament was melted in a heated extruder nozzle. Afterwards, the liquefied material was deposited on the printer bed line by line where it cooled down and hardened. After lowering the printing plate, the second layer was printed on top of the first one. The printer was not modified before use.

As illustrated in Figure 11, the positions of the LED holders of the conductive parts of the knitted fabrics were defined using a line laser, which showed the printing positions. In this way, the textile fabrics were fixed onto the heated printing bed in the desired positions. The laser was mounted mechanically on the printer frame and moved to specify the printing positions. The knitted fabric was glued on the printing bed, and the printing process was started. Alternatively, the coordinate system of the printer software can be easily transferred onto the glass printing bed using a fine permanent marker. The height of the printer nozzle is selected based on the highest adhesion between textile and 3D printed polymer. Then, the LEDs are placed in the 3D printed holders after printing finished, as illustrated in Figure 11. The shape of the selected LEDs has been used to prepare the correct holder shape in the CAD program.

Researchers [164,165] studied and investigate the adhesion properties of direct 3D printing of polymers and nanocomposites on textiles, as well as the effect of FDM printing process parameters. The adhesion forces were quantified using the innovative sample preparing method combined with the peeling standard method. The results showed that different variables of 3D printing process such as extruder temperature, platform temperature, and printing speed can have significant effects on adhesion force of polymers to fabrics during direct 3D printing.

A flexible and functional sequins developed using subtractive technology and 3D printing for embroidered wearable textile applications by Nolden [166]. The existing acrylic adhesive of the copper adhesive tape required etching and could be free from acrylic adhesive residues. This additional cleaning is complex and slightly damages the conductor structure. It can lead to defects and peeling of the copper in places, especially of fine conductor lines and angular conductor areas. Furthermore, in the additive manufacturing process, gluing with copper tape has done manually with very complex steps which automation is not possible. Moreover, for the application of wearable textile, the electronic circuit for functional sequins was attached onto fabric by stitching or embroidering via using conductive thread. The conductive thread was also sensitive to mechanical abrasion.

Zhang et al. [167] studied the direct printing of e-textile composed of core-sheath fibers by employing a 3D printer equipped with a coaxial spinneret CNTs@SF, which spun as core sheath carbon nanotubes (CNTs) (conductive core) and silk fibroin (SF) as a dielectric sheath. The resulting component was used as a triboelectric nano-generator for the harvesting of biomechanical energy from human motion and achieved a power density as high as 18 mW/m2. In addition, researchers [168,169,170,171,172,173,174], studied the performance, application, and effects of 3D printer electronic integrated in the textile substrate. Additionally, 3D printing was used for the integration of electronics with high performance. They developed an electrical interconnection. The design was based on wires that create the interconnections between the multiple layers by having these traces interconnected within additive manufactured structures.

Akbari [175] presented the fabrication and performance evaluation of 3D printed and embroidered textile-integrated passive ultra-high frequency radio frequency identification (RFID) platforms. The antenna was manufactured by 3D printing of a stretchable silver conductor directly on an elastic band. This type of 3D printed interconnection showed suitable electric performance. Recently, Ferri et al. [176] developed textile capacitive touch sensors for a successful hand gesture recognition device. The device is equipped with microchip technologies of MGC3130, which is a three-dimensional (3D) gesture recognition, motion tracking, and approach detection controller, based on integrating Gest I C.D microchips into textile by using 3D printing.

### 2.8. Stretchable Electronics

Stretchable and flexible electronic devices have attracted a significant amount of attention in recent years due to their potential applications in modern human lives. The development of flexible devices is moving forward rapidly, as the innovation of methods and manufacturing processes has greatly encouraged the research on flexible devices [177]. As a newly developed technology, flexible and stretchable electronics are emerging and achieving a great variety of applications. Because the material can be compressed, stretched, twisted, and have the flexibility to allow complex patterns, there is a high demand for application in e-textiles.

Researchers continue to develop stretchable and flexible electronic circuits by depositing stretchable electronic devices and circuits onto stretchable substrates, or by embedding them completely, in a stretchable material such as sheets of plastic or stretchable steel foil and silicones. For example, by using carbon nanotube sheets and a thermochromics silicone elastomeric, a flexible, stretchable, and breathable soft strip-shaped thermochromics resistive heater (STRH), for use in woven textile was developed by Yiming Li [178]. However, fragments of carbon nanotube are not biodegradable and poses a major health problem if it is inhaled into windpipe or lungs [179,180].

Stretchable solar cells have also been developed and used for energy harvesting [181]. Furthermore, a soft gel and textile mesh electrode was developed to create a rechargeable alkaline manganese battery with an average cell capacity of 6.5 m Ah [182]. A multiple-force sensing woven textile, used as artificial skin, was developed as coaxial structure of stretchable sensor electrodes. Although the stretchable functional textile uses only one kind of sensing unit, it can simultaneously map and quantify the mechanical deformations generated by conventional pressures, lateral strains, and flexion [183]. De Sousa Pesse et al. [184] studied and developed flexible, stretchable, washable, and wearable electronic interconnects, which had additional track on the upper and inner side of a meander shaped copper track and was encapsulated by a TPU layer to protect the electrical conducting track from mechanical abrasion.

Macdonald [185] used the latest advances in plastic-based substrates for flexible OLED (organic light-emitting diodes) integrated electronics and coupled these with recent developments in solution deposition and ink-jet printing for laying down materials and active-matrix thin-film-transistor (TFT) for the application of e-textile.

The design of metal interconnects for stretchable electronics and fabrication of elastic interconnections for stretchable electronic circuits have been conducted by different researchers [186,187,188,189,190]. They were developed by applying different fabrication technologies to embed sinuous electroplated metallic wires in the stretchable substrate or by construction of elastic point-to-point interconnections, based on 2-D spring-shaped metallic tracks.

The Center for Microelectronics Technology (CMST) at Ghent University, developed a technology to make elastic electronic devices, mainly for application onto textiles [191]. An electronic circuit was divided into functional islands encapsulated in an elastic polymer polydimethylsiloxane (PDMS, silicone), fabricated with standard technologies as shown in Figure 12.

In references [192,193,194,195], the design, development, and integration of stretchable and flexible high conductivity electronic circuits and power sources for wearable applications was studied. The physical, performance, and electromechanical characteristics of stretchable and flexible high conductivity electronic circuits was measured and analyzed. The result showed that, the research needs an improvement in mechanical durability, reliability, electrical conductivity. The improvement of stretchable electronics and fabrication by manufacturing with low cost e-textile applications has also been studied by researchers [196,197,198,199,200,201,202,203]

Cao et al. [157] fabricated a flexible, stretchable, highly conductive, and washable e-textile from conductive carbon nanotubes (CNTs) through screen-printing technology that addresses all of the concerns. They showed its application as a self-powered touch or gesture tribo-sensor for intelligent human-machine interfacing. The fabrication method is illustrated in Figure 13.

Generally, there are several critical challenges that remain for wide-scale adoption of flexible and stretchable electronics [204]. The limitation of flexible and stretchable electronics is that the device performance may be lower than conventional rigid electronics. Flexible and stretchable electronics may not be able to compete with rigid electronics device performances because when the substrate is changed from rigid silicon wafer to plastics, the device reliability would be decreased significantly. Other concerns are that there have not yet been many long-term endurance and safety tests. In addition, the effects of electrical resistance property of stretchable electronics, the capacity, and inductance behavior of stretchable electronics also changed due to the stretchable property [205]. Furthermore, the lack of coherent manufacturing technology serves as a severe challenge, specifically when the overall activity is predominantly led by the academic community 

### 2.9. Electronic Connections on Threads (E-Threads)

Studies by Vicard et al. [206], showed how a thread connection called the Diabolo process was developed. It comprises 10 stages at the wafer level and connects a die directly to external connection wires without going through a classical regular package. The die itself was protected by applying cover glue on the chips using a wafer-scale process. The connection has been established at the edge of the chip; the cover insures mechanical stability. The result was a string of chips mechanically and electrically connected to a set of wires, which is suited for further roll-to-roll processing and use in industrial processes such as weaving or extrusion. Komolafe et al. [207] also studied the integration of flexible filament circuits for e-textile applications by using copper wire, as shown in Figure 14. These processes are typically used for the applications of a LED or RFID into textile safety garments, home decoration textiles, or into composite materials.

Rein et al. [208] successfully developed integration of semiconductor diodes into textile fiber for the application of modern computation, communication, physiological monitoring, light-emitting, as well as high-bandwidth photo detecting fiber. As such, incorporating them into textile fibers can increase fabric capabilities and functions; the first soft fabric-based OLEDs encapsulated by polyurethane (PU) and poly (vinyl alcohol) (PVA) layers, via spin coating and thermal evaporation, was demonstrated. The result shows high-performance OLEDs without degrading the mechanical characteristics of the fabric itself, such as the flex stiffness and the presence of multidirectional creases. PU and PVA layers, which only slightly degrade the flex stiffness of bare fabrics due to their ductile characteristics were investigated in references [209].

Zhang et al. [210], Hsienwei Hu [211], and Qiu et al. [212] studied the integration of perovskite solar cells with a flexible fiber structure prepared, for the first time, by continuously winding an aligned multiwall carbon nano-tube sheet electrode onto a fiber electrode as illustrated in Figure 15. The result showed that the fiber-shaped perovskite solar cell exhibits an energy conversion efficiency of 3.3%, which persisted on bending. The perovskite solar cell fibers have been woven into an electronic textile to demonstrate large-scale application and further woven into flexible textile. Lee et al. [213] conductive fiber based highly sensitive textile pressure sensor are developed by employing conductive fibers manufactured by coating poly styrene block butadiene styrene (SBS) polymer on the surface of poly-phenylene tere phthalamide (Kevlar) fiber with rubber dielectric materials.

Dhawn [214] developed fiber optical systems that have been constructed by fusing different fiber optic elements together into a continuous fiber with uniform diameter. Using this approach, the light has been transmitted to the sensor efficiently by a single mode optical fiber. These showed incorporation of metallic and semiconducting nano-particles into the core of the fiber optic in-line sensors that were fabricated by coating the tip of the optical fiber with vanadium oxide and coating the tip with a protective layer of silicon dioxide.

Wang [200] studied the impact of washing on the performance of electronic integrated e-yarns, which consist of passive UHF RFID tags based on dipole antennas fabricated from copper fabric and coated with protective epoxy coating. Despite the reliability challenges related to mechanical stress, the applied epoxy coating was found to be a promising method for electro textile tags in moist conditions. Tao [91] also investigated e-yarn encapsulated with TPU (Thermoplastic polyurethane) films that were deposited by pressure under controlled temperature and pressure parameters in order to protect the conductive thread and electrical contacts, as shown in Figure 16. The reliability and washability of conductive threads and contact resistances between flexible PCB and conductive threads were found to be promising.

Ouyang et al. [88] have described washable NWF (non-woven fabric) e-textile prepared by ultrasonic nano-soldering of carbon nanotubes onto polypropylene and viscose polymer fibers, which can be used for wearable health care, as well as strain and pressure sensors. The result showed that CNT e-textiles fabricated by this method have good washability. The CNTs remain on the fiber surface even after vigorous mechanical washing in water for 40 h. The conductivity of the textile decreases slightly, and the change arises from the damage of NWF fibers rather than the rinsing away of the CNTs. The wearable computing Lab at ETH Zurich [215,216] developed a process for mounting small surface mount devices (SMD) on flexible 2-mm wide plastic strips, which contain the metal bond pads and interconnect to link components. They were woven into a textile in the weft direction in place of standard yarns. These integration methods did not meet the physical and mechanical requirements of textiles in terms of stretchability, bendability, and washability. Furthermore, the comfort was not acceptable during skin contact. The components and interconnects were left exposed at the surface of the textile and failed rapidly after washing.

Claire [217] and Paret [218] claimed the integration of chips into/onto textile material by a method known as e-thread technology. Microelectronic chips are connected to a set of two conductors behaving as an antenna, a power, or a data bus. They are encapsulated and integrated in a yarn, as shown in Figure 17. These chips can be embedded in smart packaging and allow high throughput on an assembly line. They have been integrated into textile material by a textile spool.

Recently, e-threads technology became available in three versions: firstly, with a wired sensor to track parameters such as temperature or motion, the second one with a built-in light-emitting diode (LED), and a third with an UHF (ultra-high frequency) passive RFID (Radio-frequency identification) chip and antenna to store and transmit data when interrogated. The LED version is intended for cosmetic purposes; when sewn into a garment, a car-seat cover, or some other object, it could illuminate when wired to a power source. Direct die to wire achieves a direct connection of a chip hooked onto conductor textile threads. However, the technology still has problems with washing, and the costs of e-thread is still high.

To overcome the problem [219,220,221,222,223], microelectronics was integrated into multi filament copper wire by soldering. The electronic yarns (e-yarns) contain electronics, which are fully incorporated into textile or garment production. Then, the authors have developed a semi-automated encapsulation unit to fabricate the micro-pods, since the manufacturing of e-yarn requires the creation of resin micro-pods, which protect the die and solder joints against abrasion and moisture ingress. In 2019, a prototype was made by an automated encapsulation unit and its electromechanical behavior was studied [219], as shown in Figure 18.

## 3. Outlook, Future Perspectives and Conclusions

The rapid development of functional textiles needs further creation of advanced materials, as well as new smart textiles, by integrating electronic devices in/onto the textiles substrate. Currently, several companies are working on e-textile products. Grandview research projects show that, the compound annual growth rate for E-textiles will soar over the next few years. As technology advances, there has been a growing demand for more sophisticated and e-textile products. Design and production of wearable textile issues across the industry, such as creating simple and reliable connectors for integrating electronic components or creating a washable, flexible and highly stretchable, durable, and reliable electronic component have often been resolved with proprietary solutions developed by several researchers. Early e-textile researchers, developers, and inventors have various patents in these areas, which have studied and developed a considerable barrier to entry for newcomers and has equally frustrated cost-reduction and product-improvement efforts by the existing research.

This review has clearly shown the techniques of integrating electronics into/onto textile substrates. The approaches of integrating electronics offer different advantages and disadvantages. Overall, the different approaches discussed above led to the development of a broad range of flexible, stretchable, and conformal, wearable e-textile structures with fully performance. Current challenges remain in ensuring consistent performance, while withstanding the mechanical and chemical stresses endured during wear and fabric care. For instance, resistance to washing is considered one of the primary challenges to the marketability of e-textiles [224]. In particular, the electrical performances of textile-based wearable sensors can be degraded during washing because of the mechanical deformations induced during the washing. Therefore, the durability of the washing process should be required to be further developed for use as a wearable e-textile in daily life. In addition, one of the problems discussed by Rotzler et al. [225] is the absence of standardized methods and regulation of e-textile for testing the washability of e-textiles.

The most applicable stages of integrating microelectronics into the textile are the textile adapted and textile integrated stage, but these two mechanisms have big drawbacks in flexibility and washability of the resulting e-textile material. To overcome this problem, researchers need to study the integration of ultra-thin microelectronics with or into highly flexible, stretchable textile conductive fibers or yarns. After the integration performed the conductive truck will have encapsulated by thermoplastic resins for protection of conductive truck subjected to harsh operating environments. Research has confirmed the major limitations caused by the inherent hysteresis of textile structures, which limits their application to comparative measurements and evaluations only. Therefore, researchers have taken a major step forward by proposing a third generation of electronic textiles where integrated a flexible, durable, reliable, and washable circuits are fully incorporated into yarns prior to fabric or garment production i.e. the textile based approach. It is expected that in the future the integration of electronics into textile substrate will be done by auto-mated insertion with in the textile fiber and yarn level for producing e- textile. In addition, the connection parts of the electronics into textile substrate will be fully conductive, light-weight, have high flexibility and good stretchability behavior, have chronophysiological comfort, are washable, breathable, easily wearable and can be manufactured with low cost, leading to an innovative generation of e-textile applications. At this moment several of these requirements pose challenges, with the need to be washable one of the most difficult challenge to be overcome.

In general, the integration of electronics into textile for the application of wearable e- textile shall be done in the future without affecting or influencing the final design and characteristics of the textile substrate.

## Figures and Tables

**Figure 1 materials-14-05113-f001:**
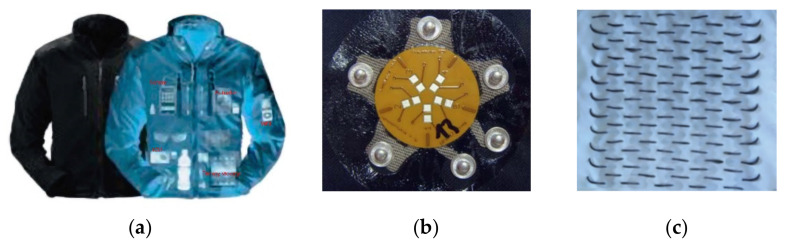
Level of electronics integration in textile: (**a**) textile-adapted, reprinted with permission from ref. [26]. Copyright Elsevier license no. 5107701406893. (**b**) textile-integrated, reprinted with permission from Taylor & Francis copy right license no. 501666551. (**c**) textile-based, reprinted with permission from [31]. Ghent University Textiles and Chemical Engineering.

**Figure 2 materials-14-05113-f002:**
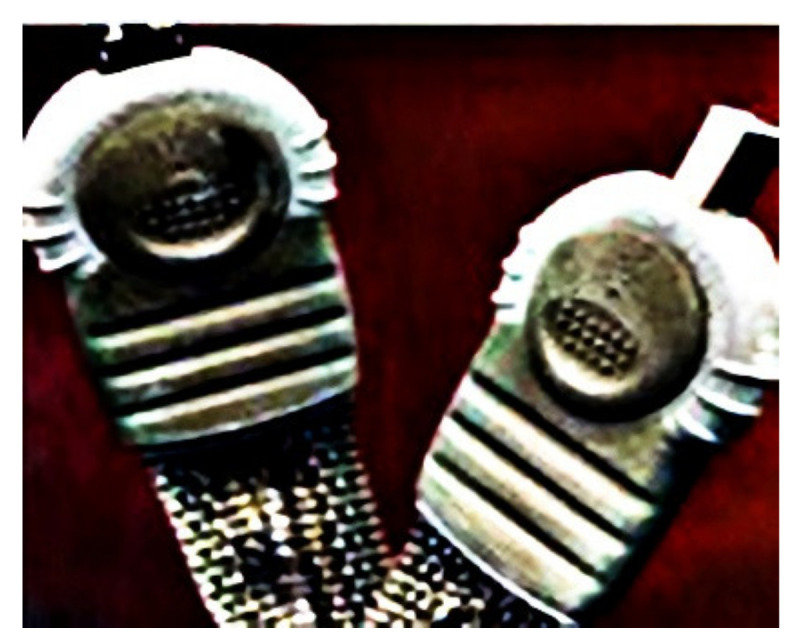
Physical attachments for electronic PCB fabric USB connector, reprinted with permission from ref. [19]. Taylor & Francis with license no. 501667190.

**Figure 3 materials-14-05113-f003:**
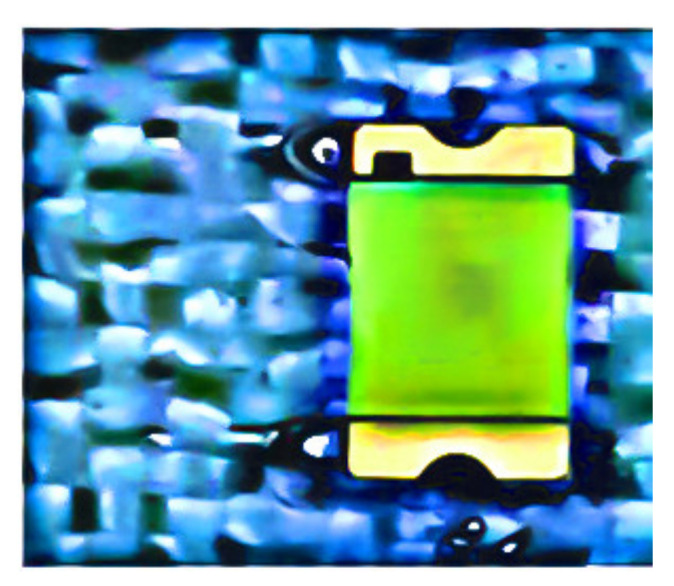
Soldering of electronics and conductive wires on textile substrate, reproduced with permission from MDPI with creative common CC by license, https://www.mdpi.com/openaccess, accessed on 13 July 2021 [91].

**Figure 4 materials-14-05113-f004:**
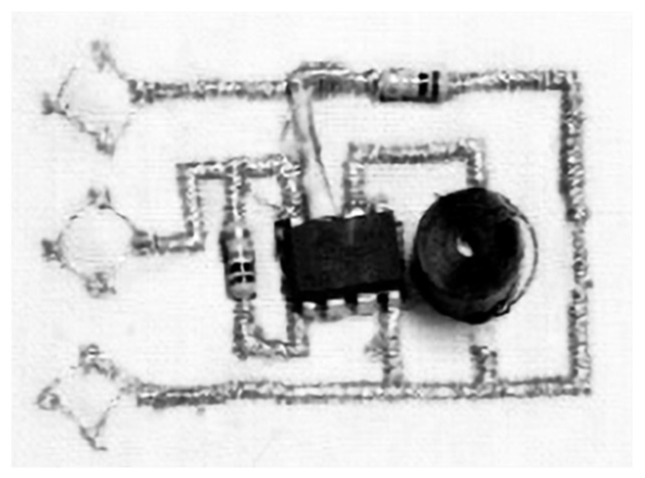
Integration and connection of microelectronics on textile substrate by embroidery, Reprinted with permission from ref. [107]. Taylor & Francis with order no. 5107740253619.

**Figure 5 materials-14-05113-f005:**
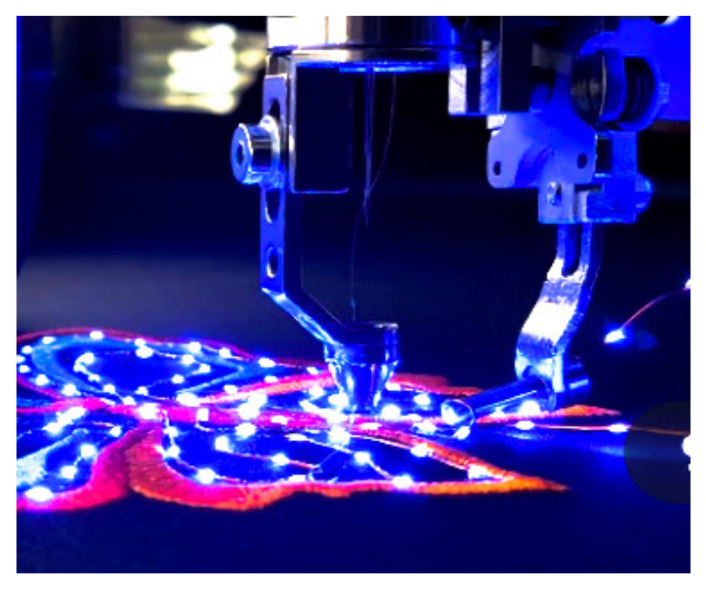
ZSK automated LEDs sequin attachment device.

**Figure 6 materials-14-05113-f006:**
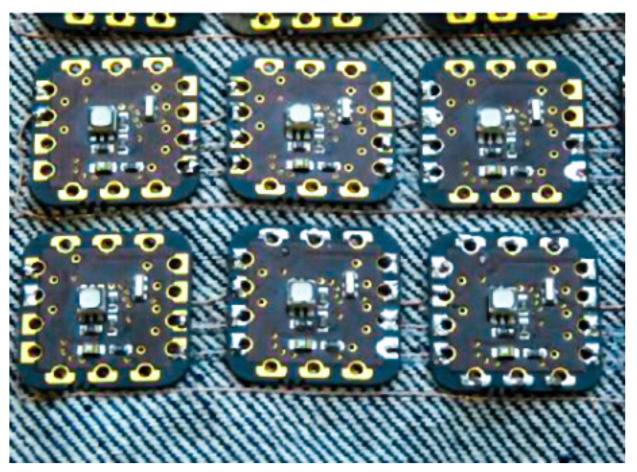
Hybrid solder and stiches of electronics on to textile, reprinted with permission from ref. [62]. Elsevier, with license no. 501667201.

**Figure 7 materials-14-05113-f007:**
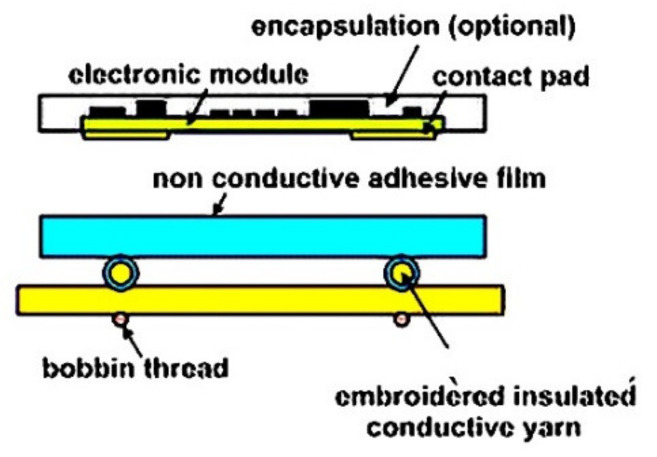
Connection of electronics by non-conductive adhesive onto textile reprinted with permission from ref. [114]. Taylor & Francis order no. 501667217.

**Figure 8 materials-14-05113-f008:**
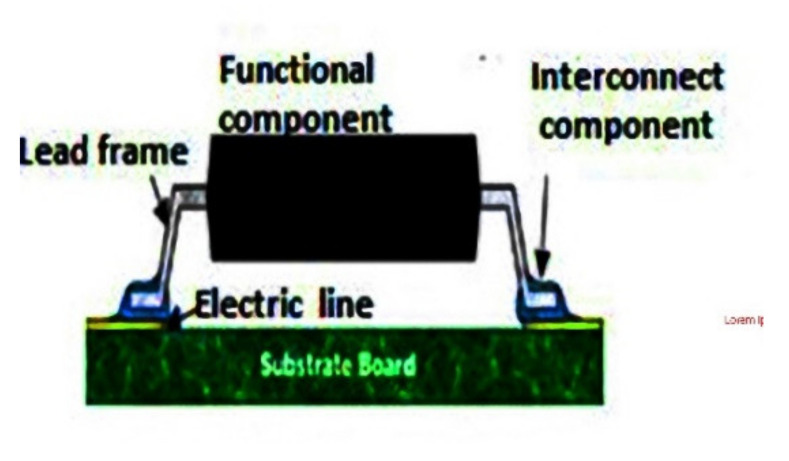
Thermoplastic bonding electronics on to textile, Reprinted with permission from ref. [122]. Elsevier, license no. 5107731320873.

**Figure 9 materials-14-05113-f009:**
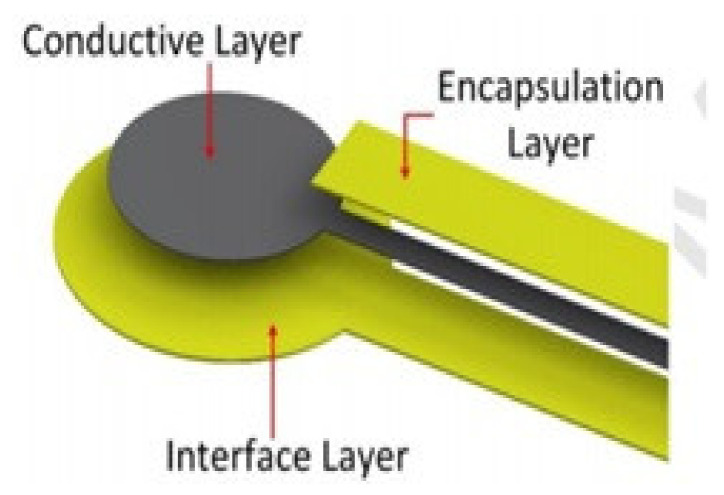
Screen printing fabrication for conductive tracks, reprinted with permission from ref. [158]. Elsevier with license no. 5107681074985.

**Figure 10 materials-14-05113-f010:**
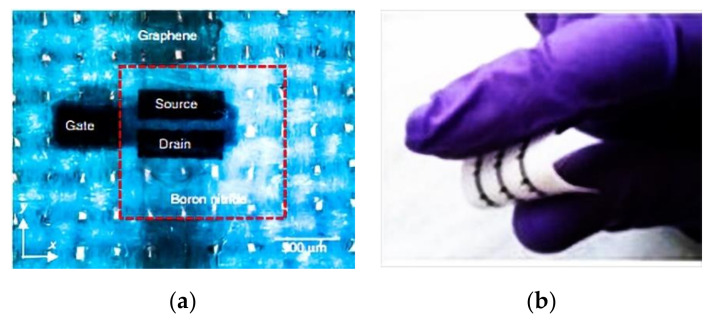
Graphene based biocompatible electronics. Optical microscopy image of the inverted FET on polyester (**a**). Image of an array of FETs on textile (**b**), reprinted with permission from ref. [156]. Springer Nature http://creativecommons.org/licenses/by/4.0/ accessed on 13 July 2021.

**Figure 11 materials-14-05113-f011:**
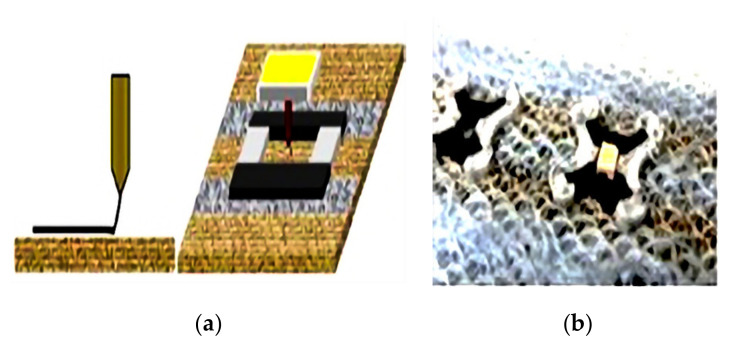
3D Printing with conductive and non-conductive polymer on knitting fabric with conductive (black) and nonconductive (grey) areas (left panel) (**a**), placing the SMD-LED in the printed holder afterwards (afterwards) (**b**), reprinted with permission from ref. [163]. Elsevier License no. 5107681074985.

**Figure 12 materials-14-05113-f012:**
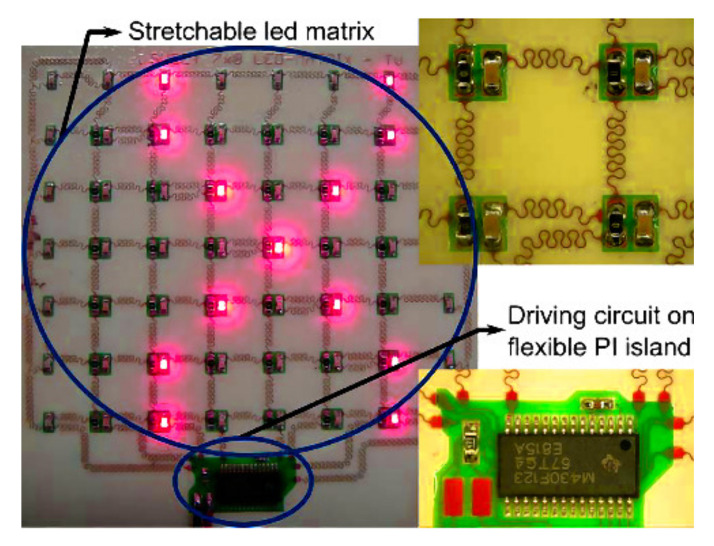
Stretchable polymer encapsulation microelectronics on textile, Reprinted with permission from ref. [191]. Taylor & Francis with order no. 501617105.

**Figure 13 materials-14-05113-f013:**
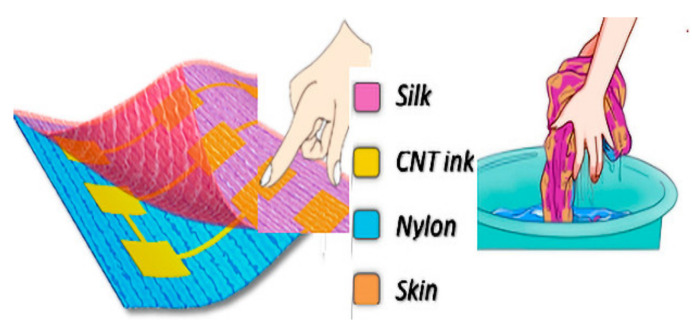
Washable flexible and stretchable electronics, reprinted with permission from ref. [157]. ACS, Copyright © 2018, American Chemical Society.

**Figure 14 materials-14-05113-f014:**
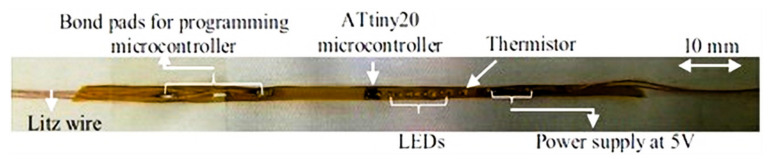
Direct die to wire connection, reprinted with permission from ref. [207]. Advanced material technology, creative common attributes, https://creativecommons.org/licenses/by/4.0/ accessed on 13 July 2021.

**Figure 15 materials-14-05113-f015:**
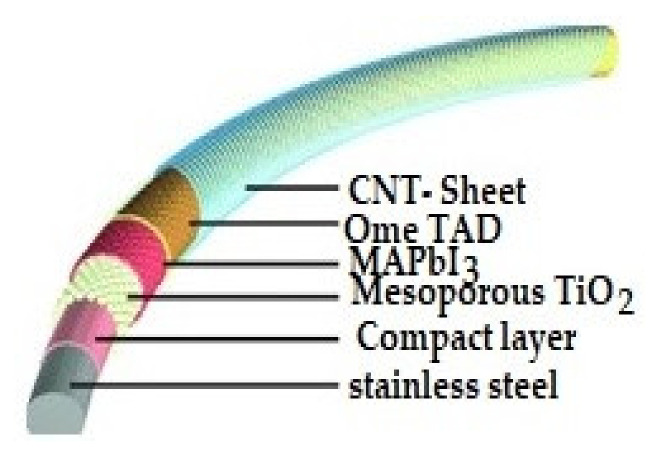
Integrating perovskite solar cells with a flexible fiber, reprinted with permission from ref. [211]. Advanced material technology, creative common attributes, htpps://creative commons.org/ licenses /by/4.0/ accessed on 13 July 2021.

**Figure 16 materials-14-05113-f016:**
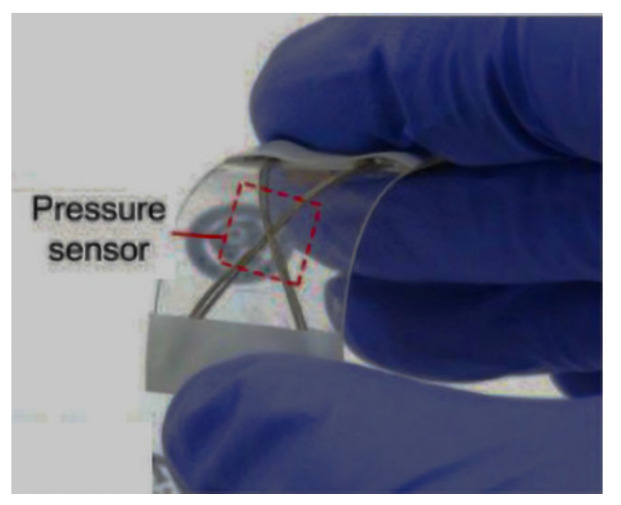
RFID chips in to flexible thread and plastic substrate, reprinted with permission from ref. [213]. Advanced material technology, creative common attributes, htpps://creative commons.org/ licenses /by/4.0/ accessed on 13 July 2021.

**Figure 17 materials-14-05113-f017:**
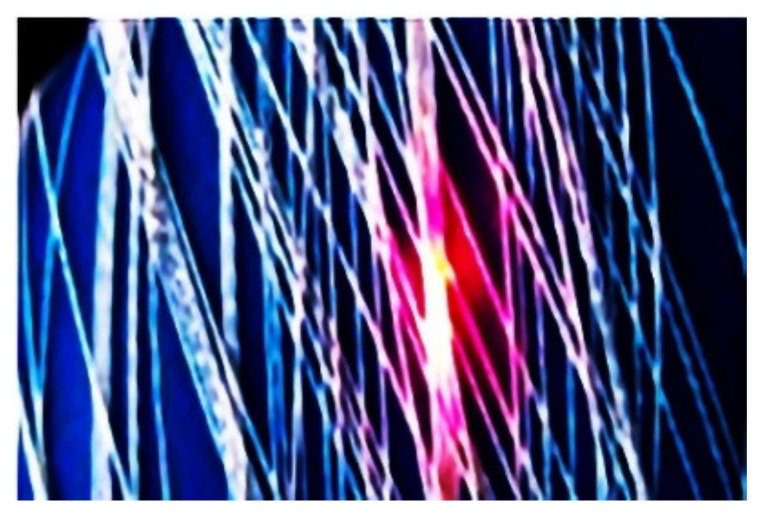
Primo1D integrate with LED [217].

**Figure 18 materials-14-05113-f018:**
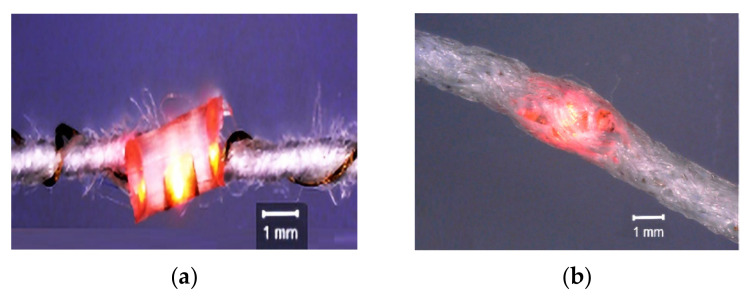
Textile yarn twisted around LED integrated coper wire (**a**). Encapsulation of microelectronics and conductive threads by polymers (**b**), reprinted with permission from ref. [219]. MDPI, https://www.mdpi.com/openaccess, accessed on 13 July 2021.

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
