# Peer review of "Review on the Integration of Microelectronics for E-Textile"

_materials, 2021, doi:10.3390/ma14175113_

Round 1

Reviewer 1 Report

The manuscript is well written as a review paper (204 references used). The integration of microelectronics for E-textile is well described.

However, the manuscript lacks an author's critical review as follows:

  1. The problem of sweat in clothing is not described. Sweat has high conductivity and can cause major problems between electronic components and conductive uninsulated yarns. The problem that arises due to atmospheric water vapor (rain, wet snow) is not mentioned either.
  2. The problem of static electricity in clothing (values up to several thousand volts) that can destroy sensitive components of e-textile is not mentioned.
  3. The problem of the influence of relative humidity on electrical resistance (e.g. cotton fiber that changes the characteristics of electronic circuits in e-textile) is not mentioned.
  4. Check the unit of measurement in line 389 "relatively high conductivity (0.2 kΩ / sq.)". Unit is Siemens, S.
  5. The use of carbon nanotubes needs to be critically reviewed. Debris from carbon nanotubes can occur specifically in a triboelectric nano generator (line 447). Such debris is not biodegradable and poses a major health problem if it is inhaled into windpipe or lungs.
  6. The terms in Figure 15 need to be translated or described in the text of the manuscript.
  7. In Chapter 2.8. Stretchable electronics, especially line 523, are facing several critical challenges. However, the fact that in addition to the electrical resistance of stretchable electronics, the capacity and inductance of electronic circuits changes is not stated. The detrimental effect of the occurrence of impedance in circuits at higher frequencies occurs and this is not mentioned.

A critical, constructive authorial review is required, as suggested in points 1-7.

Author Response

Dear Professor,

First, we appreciate your valuable comments given, and thank you for taking your precious time for the comments again.
We provided responses for each comment one by one.

  1. The response answers and corrected paragraph is highlighted by yellow color in the revised manuscript. 
  2. The resolution (Pixels) of all pictures found in the manuscripts are enhanced by Adobe Photoshop (PS). 
  3. Graphical abstract is prepared and added.

Reviewer 2 Report

1. It would be great if the technical aspects have been discussed using the science behind the particular studies.

2. Should have pick more descriptive figures and flow charts from the original studies.

Author Response

Dear Professor,

First, we appreciate your valuable comments given, and thank you for taking your precious time for the comments again.
We provided responses for each comment one by one. 
N: B 1. The response answers and corrected paragraph is highlighted by yellow color in the revised manuscript  

Reviewer 3 Report

Dear authors,

Your paper comprehensively and very well provides an overview of the different interconnection methods of electronic components into/onto textiles. At the same time, the problems related to this integration are clearly described here. The article is logically well structured and high readable. However, the present manuscript has several areas that need to be clarified or completed. I would like to point out certain aspects:

1) In the paper, I found two minor flaws, the elimination of which will help the text comprehensibility:

1a) In the caption of Figure 4, "...microelectronics on textile substrate by embroidery. reproduced..." should be followed by a comma instead of a full stop "..microelectronics on the textile substrate by embroidery, reproduced..." and "no 5107740253619" should be followed by a full stop "no. 5107740253619".

1b)The captions such as “Schicht or Edestahl” in Figure 15 are in German; for a better understanding of the text, it would be helpful to translate these captions into English.

2) It should be considered whether a graphical abstract, which is now quite common in review articles, would help readers quickly understand the take-home message of the paper, and at the same time, could be beneficial both in terms of views and citations of your paper.

3) On lines 57 to 59, you state that Lina Rambausek discussed the five different levels of textile integration in her Ph.D. thesis and that Bosowski et al. generally studied the three category levels of the integration of electronic components and circuits. However, a working group (WG31) was assembled in 2007 in the framework of CEN to start setting up standardization of smart textiles, within the Technical Committee 248 (CEN/TC 248 WG31), which published the Technical Report 16298:2011 in 2011. The committee during the revision of the CEN/TR 16298 defined the following four different levels of integration an electronic component or device is integrated onto or into a textile material or textile product:

a)Integration level 1 (removable): electronic device is added in a way that it is removable without destroying the product, e.g. by means of a pocket, touch-and-close fastener, push button.

  1. b) Integration level 2 (attached): electronic device is attached to textile in a way that it is not removable without destroying the product, e.g. by means of stitching, welding or glueing the device onto the textile.

  1. c) Integration level 3 (mixed solution): electronic device consists of one or more components made of textiles or through textile finishing and combined with permanently or non-permanently attached electronic components, e.g. an LED lamp attached to a conductive track woven into a fabric.

  1. d) Integration level 4 (full textile solution): all components of the electronic device are made of textile or textile finishing.

This division into 4 Integration levels is very close to reality, can you please add it to your article.

4) It would be useful if this paper also addressed the issue of the lack of standards and regulation for e-textiles, which to a certain extent hinders commercialization.

5) In section "2.2 Soldering" you clearly discuss welding as a method of interconnections between microelectronics and conductive textile material. However, nowhere in the text is the essential information that most commercially available conductive threads based on silver-metallized polyamides cannot be soldered. Conversely, yarns that are prepared by plying non-conductive filaments with metallic microwires, which are not mentioned at all in your article, can be soldered very well.

6) Information about thermoplastic ultrasonic welding and resistance welding of conductive yarns, which appear to be very promising alternatives to existing interconnection methods, completely missing in your review paper. Moreover, these methods are very fast, easily scalable, and compatible with textile processes.  Could you please add this information to your article.

7) On line 299, you mentioned Madeira high conductive thread (HC12) as the only representative thread for the integration and interconnection of electronic components onto textile substrate by embroidering. This part of the article could be supplemented with other commercially available threads such as Amann SilverTech, Volt Smart Yarns, or Clevertex threads. As Maidera HC12 is relatively thick, which is not quite suitable for high-resolution embroidery and does not have a very high resistance to washing cycles.

8) Regarding 3D printing, which is discussed in chapter "2.7 Three-dimensional (3D) printing", have you found information in the cited papers regarding the adhesion of a 3D printed plastic material to textile fabric. And whether any peel tests have been done in their research and the automated wash resistance of such directly 3D printed structures was tested during the cyclic washing tests.

9) In the section entitled "3. Outlook, future perspectives and conclusion" you state the claim that for textile-integrated levels of integration there are big drawbacks in flexibility and washability. Could you please provide more justification and support for this claim. Do you mean that all the technologies and methods for the textile-integrated level have been exhausted and that there cannot be a significant increase in quality, long-term reliability and resistance to mechanical, chemical, thermal stresses and wash cycles?

Conversely, you suggest the integration of microelectronics with or into textile fibers or yarns as a solution to overcome the above-mentioned problems. Could you please suggest ways in which the individual conductive tracks of such systems will be interconnected on the fiber level so that they exhibit a high degree of flexibility, stability and resistance to washing cycles?

10) The resolution of the pictures could be higher. It is a bit better on the portal Preprints (www.preprints.org), where this paper was posted on 21 July 2021. See at https://www.preprints.org/manuscript/202107.0388/v2/download. Maybe it is due to compression, but please check it.

Author Response

Dear Professor
Warm greetings!
First, we appreciate your valuable comments given, and thank you for taking your precious time for the comments again.
We provided responses for each comment one by one. There were three reviewers who ask different questions and the response answer is given in the revised manuscript and referees Comments.
N: B 1. The response answers and corrected paragraph is highlighted by yellow color in the revised manuscript  
      2. The resolution (Pixels) of all pictures found in the manuscripts are enhanced by Adobe Photoshop (PS)  
      3.  Graphical abstract is prepared and added.

Round 2

Reviewer 1 Report

The authors modified the manuscript according to the review.